# Viscoelastic Polyurethane Foam with Keratin and Flame-Retardant Additives

**DOI:** 10.3390/polym13091380

**Published:** 2021-04-23

**Authors:** Krystyna Wrześniewska-Tosik, Joanna Ryszkowska, Tomasz Mik, Ewa Wesołowska, Tomasz Kowalewski, Michalina Pałczyńska, Damian Walisiak, Monika Auguścik Królikowska, Milena Leszczyńska, Krzysztof Niezgoda, Kamila Sałasińska

**Affiliations:** 1Łukasiewicz Research Network, Institute of Biopolymers and Chemical Fibres, ul. Skłodowskiej-Curie 19/27, 90-570 Łódź, Poland; zdp@ibwch.lodz.pl (T.M.); e.wesolowska@ibwch.lodz.pl (E.W.); t.kowalewski@ibwch.lodz.pl (T.K.); protein@ibwch.lodz.pl (M.P.); d.walisiak@ibwch.lodz.pl (D.W.); 2Faculty of Materials Science, Warsaw University of Technology, Woloska 141, 02-507 Warszawa, Poland; joanna.ryszkowska@pw.edu.pl (J.R.); monika.auguscik.dokt@pw.edu.pl (M.A.K.); Milena.leszczyska.dokt@pw.edu.pl (M.L.); krzysztof.jerzy.niezgoda@gmail.com (K.N.); 3Department of Chemical, Biological and Aerosol Hazards, Central Institute for Labor Protection, National Research Institute, Czerniakowska 16, 00-701 Warsaw, Poland; kasal@ciop.pl

**Keywords:** viscoelastic foams, flammability, thermal analysis, physical properties, mechanical properties

## Abstract

Viscoelastic polyurethane (VEPUR) foams with increased thermal resistance are presented in this article. VEPUR foams were manufactured with the use of various types of flame retardant additives and keratin fibers. The structure of the modified foams was determined by spectrophotometric-(FTIR), thermal-(DSC), and thermogravimetric (TGA) analyses as well as by scanning electron microscopy (SEM). We also assessed the fire resistance, hardness, and comfort coefficient (SAG factor). It was found that the use of keratin filler and flame retardant additives changed the foams’ structure and properties as well as their burning behavior. The highest fire resistance was achieved for foams containing keratin and expanding graphite, for which the reduction in heat release rate (HRR) compared to VEPUR foams reached 75%.

## 1. Introduction

Viscoelastic polyurethane (VEPUR) foams or memory foams are foams designed to improve the comfort of astronauts at take-off and landing during space flights [1]. Currently, these foams are often used to manufacture: Mattresses and pillows that provide comfort for their users during sleep, elements supporting the treatment process used in medicine, or many vibration damping elements used in the automotive industry [1]. Viscoelastic foams reduce the possibility of bedsore formation in long-term patients, which makes they are used in the care of the elderly [2,3,4,5]. The research results of these foams indicate the possibility of extending the direction of their applications, but due to the main directions of their application in the automotive industry and in medicine, including long-term care, their high resistance to fire is required [6]. Polyurethane foams burn rapidly releasing large amounts of heat. A lot of toxic gases and smoke are also emitted during the combustion [7].

Viscoelastic foams belong to the group of flexible polyurethane foams (FPUF) and their thermal decomposition is similar. It is a two-step process: Each of the stages of thermal decomposition occurs at a different temperature, which during a fire is related to the formation of two peaks of heat release [8,9,10]. In the case of flexible foams, their ignition and flame combustion occurs after a short time and constitutes a significant fire hazard. To reduce this risk, flame retardant additives are used in the foam manufacture. The effectiveness of such additives has been confirmed in many review articles [11,12,13,14,15,16,17], and up-to-date information on such additives are presented in the work of the United States Environmental Protection Agency [18] and the works of many authors [19,20,21,22,23,24].

Many flame retardants cannot be used in the USA and EU [25,26,27,28], therefore, new compounds are looking for to eliminate these hazards.

To manufacture flexible flame retardant foams, halogen compounds, phosphorus, melamine, or a combination thereof are used. Other groups of organic and inorganic compounds, including reactive and nonreactive compounds, have also found application [18,19,20,21].

One of the most effective groups of additives influencing the flammability of PUR foams are inorganic fillers. During the degradation of foams with such additives, a stable organic-inorganic layer is formed on their surface, limiting the concentration of gases generated during decomposition, which also acts as a barrier to their diffusion [29]. Such a solution has proven successful in the production of aircraft seats, in which expanding graphite (EG) was used [30] and is now a very popular flame retardant [31,32,33]. Expandable graphite is a material resulting from the exfoliation of graphite crystals intercalated with sulfur, phosphorus, or fluorine atoms [34]. Depending on the type of compounds used for graphite intercalation, EG is formed, which has a different effect on the thermal resistance of polyurethane foams [31].

To increase the effectiveness of EG, other compounds are additionally introduced in the production of PUR foams: i.e., aluminum hypophosphite [35], ammonium polyphosphate [36] or pentaerythritol phosphate [37]. The group of inorganic flame-retardant fillers also includes magnesium hydroxide (MTH) and aluminum hydroxide (ATH) [38,39].

The group of inorganic compounds that change the flammability of polymers also includes metal oxides: An example of such an oxide is zinc oxide (ZnO) [40,41]. The effectiveness of these compounds is related to their use in the form of powders with nanometric particle sizes [40,41]. The use of metal oxides in the form of nanofillers may affect the kinetics of the reaction in foam formation [42].

An interesting additive that reduces the flammability of foams are phosphorus compounds, i.e., for example, Fyrol ™ PNX LE (Ltd Tel Aviv, Israel) with 19 wt% of phosphorus. The results of the research of viscoelastic foams with the addition of this Fyrol were presented by Zieleniewska and co-authors [43], as a result of them it was found that the introduction of 12% Fyrol significantly increases the isocyanate index (IO) to 23% and reduces the maximum heat release rate (pHRR) from approx. 440 kW/m^2^ (for the reference foam) to approx. 90 kW/m^2^ (for foam with 12% Fyrol) [43].

In recent years, a significant increase in the use of natural fillers for the production of polymer composites has been observed [44]. One of the less frequently used fillers for the production of polymer composites are fibers and particles of keratin obtained from poultry feathers [45]. Keratin fibers were used to produce FPUF. Based on the characteristics of these materials, it was found that the use of 10 wt% of keratin fibers caused the increase of the oxygen index from 19.8 for the reference foam to 22.8%. Moreover, the reduction of the effect of falling drops of burning material was also observed [46].

The aim of the article was VEPUR foams with increased thermal resistance, made with the use of various types of flame retardant additives and keratin fibers. A few research methods, such as thermogravimetric analysis, differential scanning calorimetry, and cone calorimetry tests, were applied to define the impact of FR on the thermal stability and burning behavior of modified VEPUR foams. A microstructure analysis complemented the influence of the keratin fibers and fire retardant systems on foams’ performance.

## 2. Materials and Methods

Component A (FAMPUR, Bydgoszcz, Poland) and Component B-ONGRONAT^®^ TR 4040 (BorsodChem, Kazincbarcika, Hungary) were used for the synthesis of PUR foams. Component A is the know-how of FAMPUR.

The foams were made using the following additives:Keratin filler with particle size 0.01–0.04 mm (K), (Łukasiewicz Research Network—Institute of Biopolymers and Chemical Fibers, Łódź, Poland).Fyrol PNX (F)—ICL Industrial Products Ltd., Tel-Aviv, Israel), an alkyl phosphate.Expandable Graphite (EG)—Sinograf SA, Toruń, Poland), the particle size 0.5 mm, expansion 250 mL/g, starting expansion temperature 220 °C.Aluminum hydroxide, MARTINAL (ATH)—(Albemarle, Charlotte, NC, USA), the particle size 10 µm.Magnesium hydroxide, MAGNIFIN (MTH)—(Albemarle, Charlotte, NC, USA), the particle size 20 µm.Zinc oxide (ZO—(Unipress Institute of High Pressure Physics, Warsaw, Poland), the particle sizes 70 nm.Ammonium polyphosphate, Exolit AP 422 (APP)—(Clariant, Muttenz, Switzerland).

### 2.1. Preparation of Keratin Feather

Breaking up of the feathers: Purified, wet feathers were disintegrated several times: By cutting on a guillotine about 1.5 h and by grinding on a disk mill, SproutWaldorom; slot 0.3 mm, time ca. 4.5 h, by grinding on a disk mill, SproutWaldorom; slot 0.1 mm, time ca. 8 h. The material was then centrifuged and dried in a lacquer dryer with 30 sieves for 3 h at 80 °C, then ground in a disintegrator (KMW, 1953 Karlstad Sweden) for 10 min, and next in a ball mill for 8 h.

### 2.2. Preparation of Foams

The foams were made by one-step method. Flame-reducing additives (flame retardant—FR) were introduced into the polyol component (the amount of additives was calculated per 100 g polyol). Foam synthesis was carried out at an ambient temperature of 20 °C. Polyols and modifying additives (polyol masterbatch) were mixed with a high speed stirrer at 800 rpm for 60 s. Then the K was introduced and the whole system was mixed with the use of a mechanical stirrer at 200 rpm for 30 s. The foams were made at the isocyanate index of 80. The description of the foam composition and synthesis process parameters is presented in Table 1.

## 3. Experimental

The apparent density of the foams (d) was assessed on the basis of volume measurements with an accuracy of 0.01 mm and the mass of samples with an accuracy of 0.001 g. The test was carried out in accordance with PN-EN ISO 845: 2010.

The microstructure of the keratin particles and composites was carried out after spraying the samples with a mix of palladium and gold layers. A scanning electron was used to conduct the observations, using a 15 keV voltage microscope (SEM) TM3000 from Hitachi Group, Tokyo, Japan, was used.

Spectrophotometer Nicolet 6700 (Thermo Electron Scientific, Waltham, MA, USA) with ATR (total reflection suppressed) was used to analyze the chemical composition of the filler and composites. Each sample was scanned 64 times in the 400–4000 cm^−1^ wavelength range. Analyses of the obtained spectra were carried out with the OMNIC 8.2.0.387 Thermo Fisher Scientific Inc. software.( Waltham, MA, USA).

The analysis of the course of thermal degradation of keratin fibers and composites was carried out with application Thermogravimetric analysis (TGA) using TGA Q500 (TA Instruments, Lukens Dr, New Castle, DE, USA). Tested samples weighing 10 ± 1 mg, placed in platinum dishes, were heated at a rate of 10 °C/min from room temperature to 1000 °C under nitrogen and under air atmosphere. The results were analyzed using Universal Analysis 2000 software version 4.7A (TA Instruments Inc., New Castle, DE, USA).

Thermal analysis of the samples was performed using Differential scanning calorimetry (DSC) DSC Q1000 (TA Instruments, Lukens Dr, New Castle, DE, USA). Measurements were made in a helium atmosphere in hermetic aluminum pans. Samples of approximately 5 mg were heated at a rate of 10 °C/min from −80 to 210 °C.

The heat released during combustion, the start and end times of burning, and the weight loss of the sample was determined with the MLC cone microcalorimeter (Fire Testing Technology Ltd., East Grinstead, UK). The samples with dimensions of 100 × 100 × 10 mm were tested by heat flux at 25 kW/m^2^.

A mechanical compression test was carried out in accordance with the PN-EN ISO 3386-1: 2000 standard on a Zwick Z005 testing machine (Zwick Roell, Ulm, Germany). Based on the results of the fourth compression, the foam comfort factor was determined (stress at 65% load/stress at 25% load) and stress when loading the samples at 40% deformation, Compression Load Deflection CLD40, which was taken as a foam hardness parameter.

Compressive set (CS) deformation was determined on 50 × 50 × 50 mm samples in the direction of foam growth. Foams were compressed by 50%, 75%, and 90% of the original height in metal covers. Then they were placed in an oven at 70 °C for 22 h. After this time, the samples were removed from the metal covers and after 0.5 h at room temperature their height was again measured. The percentage loss in height of the samples was calculated, thus determining the compression set: CS (22 h, 70 °C, 50%), CS (22 h, 70 °C, 75%), CS (22 h, 70 °C, 90%). The test was carried out according to the standard PN-EN ISO 1856:2018-09.

## 4. Results and Discussion

### 4.1. Characteristics of Keratin Fibers

Figure 1 shows the image of keratin fibers used for foam modification. Based on the analysis of the fiber images, the aspect ratio of fibers was calculated as 2.09.

The fibers were also characterized by thermal analytical methods: DSC and TGA. Figure 2 presents the results of the DSC and TGA analysis of keratin fibers. The degradation process was analyzed using the curve of differential mass change DTG.

The DSC thermogram shows two endothermic peaks related to water evaporation, with a minimum of about 60 °C and 160 °C, on the DTG curve, they correspond to peaks ending at a temperature of approximately 210 °C with a 12% water loss. Similar results are presented in the work of Brebu and Spiridon [47] and the work of Senoza et al. [48]. The third peak on the DSC curve relates to the melting of protein compositions bridged by di-sulfides [48]. Keratin fiber degraded at a temperature above 236 °C [47]. Next, peaks in the DTG curves are related to the degradation of the disappearance of disulfide bridges in keratin.

The chemical structure of K fibers was analyzed using infrared spectroscopy, the FTIR spectrum is shown in Figure 3. The results of this analysis are summarized in Table 2.

### 4.2. Characteristics of Foams

On the basis of previous research on flexible [40] and viscoelastic [37] foams, it was proposed that the addition of keratin should be used to reduce the flammability of VEPUR foams. Keratin fibers are flame retardant materials, they may contain sulfur in the amount of approx. 3 ÷ 5% by weight, therefore it was assumed that their addition would increase the thermal resistance of VEPUR produced with the use of various types of flame retardant additives. It was proposed to incorporate 10 g K per 100 g of polyol component into the foams.

In addition to keratin fibers, to reduce the flammability of VEPUR, inorganic additives such as aluminum hydroxide, magnesium hydroxide, and expandable graphite were used; from the group of nitrogen and phosphorus compounds—ammonium polyphosphate, and from the group of metal oxides—the zinc oxide nano-powder.

A series of foams with the mentioned additives and Fyrol PNX was also made. As shown in the study by Zieleniewska et al. [43] with the use of 12% Fyrol, the rising time and gel time significantly increased, therefore in this study a reduction in the amount of Fyrol used was proposed. In the case of the analyzed materials, the characteristic times achieved during the synthesis did not differ significantly.

#### 4.2.1. Chemical Constitution

To analyze changes in the chemical constitution of foams containing K and Fyrol, ATR-FTIR analysis was performed (Figure 3).

Figure 3 shows the ATR-FTIR spectrum of unmodified VE foam and VE foam with keratin fibers and Fyrol. In the range of wave number 3300 ÷ 3500 cm^−1^ appear bands from reaction substrates, while in the range 3200 ÷ 3600 cm^−1^ there are bands derived from the stretching vibration of N-H bond. In the analyzed foams, in this wave number range, there is a multiplet band containing signals from stretching vibration, symmetrical, and asymmetrical, assigned to N-H bonds and from the group–OH from the polyol component unbound in polyurethane macromolecules.

In the range of wave number 2800 ÷ 3000 cm^−1^, there are bands from of stretching vibrations within a group–CH in the soft segments formed from polyols [56,57,58].

The foams were made at the isocyanate index of 0.8, therefore the spectra do not show signals around 2270 cm^−1^ attributable to-NCO bond of the unreacted isocyanate.

In all analyzed samples were also observed bands derived from bond vibrations of C = O (1728, 1711, 1664, 1658, and 1650 cm^−1^), C = C from aromatic ring (1597 cm^−1^) bending and deformation vibrations derived from N–H bonds within HNC = O (1534 and 1512 cm^−1^), H3C–C (1450 cm^−1^), –O–CH2 (1411 cm^−1^) and νasym CO/sym within the group–NCO–O (1230 and 921 cm^−1^) in–C–O–C–group [56]. In the range around 756 cm^−1^, the band represents a C–H bond from the aromatic ring.

Polyurethane macromolecules are built up of two different segments: Flexible and rigid. Rigid segments are built from urethane and urea bonds. The introduction of flame retardant additives may cause the change in the structure of the rigid segments. The introduction of keratin to the VE foam causes clear changes in the ATR-FTIR spectrum of the VE + 10 K foam in the wave number range of 1640 ÷ 1750 cm^−1^ related to the vibrations of carbonyl groups in the urea and urethane bonds of the foam (Figure 4).

After introducing keratin in the substrate mixture, isocyanate groups may react with NH2 groups on the keratin surface, as a result of this reaction, urea bonds are formed.

A scheme of such a reaction was presented in the work of Członka et al. [59]. In addition, when other additives are used, the water introduced with them may react with the NCO groups. This reaction may result in the formation of urea bonds and a by-product in the form of carbon dioxide.

To assess the change in the share of urea bonds in the structure of rigid segments selected VEPUR foams with keratin, an analysis of multiplet bands in the range of carbonyl vibrations was performed (Figure 4) [60,61].

To estimate the signal strength, peak modelling of the infrared active carbonyl bands was carried out using Gaussian curve-fitting method software OMNIC 7.3. The carbonyl absorption bands were deconvoluted into component bands. Peak areas of these bands were measured, the contribution of urea (*U*_1_) group calculated using Equation (1), and the content of urethane (*U*_2_) group calculated using Equation (2).
(1)$$U1_{\;}=\frac{\sum \;A_{1i}}{\sum \;A_{i}}$$
(2)$$U2_{\;}=\frac{\sum A_{2i}}{\sum \;A_{i}}$$
where: *U*_1_ and *U*_2_ are the contributions of urea (1) and urethane (2) groups; *A*_1*i*_, *A*_2*i*_ are the absorbance of carbonyl groups of urea (1) and urethane (2) groups, respectively and *A_i_* is the absorbance of carbonyl groups.

The carbonyl hydrogen bonding index *R_C_*
_= *O*_ was calculated using Equation (3). Moreover, the degree of phase separation *DPS* was obtained through Equation (4).
(3)$$R_{C=O}=\frac{A_{B1}+A_{B2}}{A_{F1}+A_{F2}}$$
(4)$$\mathrm{DPS}=\frac{R_{C=O}}{R_{C=O}+1}$$
where: *A_B_*_1_, *A_B_*_2_—the respective surface areas of bands with vibrations related to the hydrogen bond of the amine or carbonyl groups of urea (1) and urethane (2) bonding. *A_F_*_1_, *A_F_*_2_—the respective surface areas of bands with vibrations unrelated to the hydrogen bond of the amine or carbonyl groups of urea (1) and urethane (2) bonding.

Results of the characteristics of chemical structure of the hard phase of VEPUR foams are presented in Table 3.

The data presented in Table 3 show that the introduction of keratin filler promotes an increase in the share of urea bonds in the rigid segments of the tested VEPUR foams.

Compounds with an active hydrogen atom in the amino group are characterized by a much greater reactivity in the reaction with isocyanates than compounds containing OH groups [62], first urea groups are formed, and then urethane groups.

When the amount of isocyanate in the reaction medium does not change and the amount of amine groups in the rigid foam segments increases, the proportion of urea bonds increases.

In the VE + 10 K + 5 F foam, a reduction in the share of urea bonds in the rigid segments of this foam was observed. Fyrol does not contain functional groups but can change the pH of the reaction medium. Increasing the acidity affects reduction of the rate of reaction of isocyanate groups with hydroxyl groups and amine [58,63]. Probably the effect of Fyrol on the reduction of the rate of reaction with amino groups is greater than on the rate of reaction with hydroxyl groups. As a result, the share of urea groups in the rigid segments of VE + 10 K + 5 F foam is lower than in the VE + 10 K foam.

Based on the ATR-FTIR analysis of foams containing EG: For VE + 10 K + 10 EG and VE + 10 K + 10 EG + 5 F, it was found that after its application, the share of urea bonds in the foams increased compared to the foams VE + 10 K and VE +10 K + 5 F, respectively.

Probably the reason for such a change was introducing additional water into the reaction medium together with the filler. The use of Fyrol in the VE + 10 K + 10 EG + 5 F foam resulted in a reduction in the share of urea bonds in the rigid segments of this foam compared to the VE + 10 K + 10 EG foam. Changes in the share of urea bonds in the foams are shown in Figure 5.

In the remaining foams, it was observed that the introduction of Fyrol causes a reduction in the content of urethane bonds.

The use of keratin filler promotes the increase the proportion of hydrogen bonds and the degree of phase separation in foams.

#### 4.2.2. Morphology

Figure 6 shows an image of the cross section of the foam perpendicular to the direction of foam rise. The image of this foam shows oval-shaped pores of various sizes in the range of 30 ÷ 140 µm. The foam VE has an open cell structure. The pores are interconnected by many cell windows, the sizes of which are in the range of 3÷30 µm. Cell wall thickness is approximately 1 µm.

After the introduction of additives changing the flammability of the VE foam, the cell structure of the foams’ changes. The images of these modified foams are shown in Figure 7.

The image of the VE + 10 K foam (Figure 7a) indicates that the incorporation of keratin causes the pore size decrease, but the cell wall thickness is increased compared to the structure of the VE foam. The shape of many pores changes from ovals to polygons, parts of the pore walls are destroyed. The introduction of Fyrol to the VE + 10 K + 5 F foam causes further changes in the structure of its pores. The cell wall thickness is decreased compared to the VE + 10 K foam, much more pore walls are destroyed. Pores become irregular in shape. To explain the reason of these changes, the observation of the cell walls of the foams after modification was performed (Figure 8). It was observed that in the pore wall of the foam with keratin, there are visible keratin particles regularly distributed in the VEPUR foams mass (Figure 8a).

The pores of the cell walls of the VE + 10 K + 5 F foam contain a significant amount of agglomerates of keratin particles (Figure 8b). Keratin contains polar groups which favor the formation of strong interfacial interactions, probably the introduction of Fyrol promotes the formation of interaction between the keratin particles and the formation of their agglomerates.

In ATH and MTH foams (VE + 10 K + 10 ATH, VE + 10 K + 10 MTH), foam with ZO nanoparticles: VE + 10 K + 10 ZO and EG foam: VE + 10 K + 10 EG, the pore structure changes slightly compared to VE foam + 10 K, a similar conclusion can be drawn by analyzing the images of the structure of VE + 10 K + 10 ATH + 5 F, VE + 10 K + 10 MTH + 5 F, VE + 10 K + 10 ZO + 5 F and VE + 10 K + 10 EG + 5 F foams compared to VE + 10 K + foam 5 F.

In VE + 10 K + 10 APP and VE + 10 K + 10 APP + 5 F foams, the pore wall thickness is greater than in other foams. On the base of SEM images of foam structures, the pore schemes typical for foams: VE + 10 K + 10FR (scheme A) and for foams of the type: VE + 10 K + 10FR + 5 F (scheme B) were proposed (Figure 9).

#### 4.2.3. Apparent Density

Apparent densities of the obtained materials are presented in Table 4. Comparing with the pure VE, the apparent density of foams VE + 10 K and VE + 10 K + 5 F shows increase. When the next FR is introduced to the mixture, the apparent density of the foams increases in relation to the VE + 10 K and VE + 10 K + 5 F foams. Probably the apparent density increased after adding FR, because the density of EG was higher than that of the neat VEPUR. Densities of ATH, MTH are 2.4 g/cm^3^, EG—about 2.0 g/cm^3^, and APP—1.9 g/cm^3^. Such changes in apparent density were observed in rigid foams with EG [64] and with EG and APP [65]. The apparent density of most of the foams with the addition of Fyrol is significantly higher, except for the foams with APP and ZO. The level of apparent density of the analyzed foams is similar to the changes observed in the foams tested by Zieleniewska et al. [58]. The introduction of Fyrol causes the change of reactivity of substrate blends, what consequently causes considerable elongation of the synthesis of foams [59]. SEM images of the foams suggest that the use of Fyrol causes significant changes in the structure of the macropores, a larger number of interruptions in the pore walls appears (Figure 9). Similar changes in the structure were observed in the foams tested by Zieleniewska et al. [58].

#### 4.2.4. Thermal Analysis

The change in the chemical structure of foams caused by the introduction of keratin, a flame retardant, and Fyrol, may cause a change in the characteristic temperatures and features of transformations occurring during their heating. To determine the level of these changes, DSC analysis was performed. Selected DSC thermograms are shown in Figure 10. Based on the DSC thermograms, Tgs1 and Tgs2 were determined—the glass transition temperature in the first and second heating cycle, respectively, Tt—temperature related to the transformation disorder order and enthalpy of this transformation—ΔHt (Figure 10a,b) and the specific heat—Δcp * in the scope of glass transition transformation. The method of calculating this value is presented in [66]. The results of the DSC thermogram analysis are summarized in Table 5.

Viscoelastic foams are foams composed of a soft phase, a soft phase with rigid segments dispersed in it, and a hard phase. In the case of the tested foams, the soft phase is described by the glass transition, the soft phase with stiff segments dispersed in it, the endothermic order disorder change (Figure 10a). In the second heating cycle, only the glass transition temperature of the soft phase is marked (Figure 10b), which indicates that in the analyzed VEPUR foams the stiff segments after the heating process remain dispersed in the soft phase. The glass transition temperature of the soft phase (Tgs1) of the tested foams varies in the range of −58.6 ÷ −61.0 °C. All analyzed materials have a soft phase with the same chemical structure, therefore their glass transition temperature slightly differs. This indicates that the incorporation of flame retardant does not alter significantly the mobility of the soft phase [67]. The temperature of the glass transition determined during the second heating cycle (Tgs2) is slightly higher than (Tgs1). This indicates that the degree of dispersion of the rigid segments in the soft phase changes slightly during heating. Specific heat capacity in Tgs1 (Δcp *), normalized to polymer content is more varied (Figure 11).

The content of keratin and flame retardant may be responsible for the change in the thermal effect associated with the glass transition of the soft phase. Similar changes in the heat effect after the filler introduction were observed by Raftopoulos et al. [68]. The modification of the VE + 10 K foam with Fyrol causes that the Δcp * of the VE + 10 K + 5 F foam is much lower than the VE and VE + 10 K foams. Fyrol is an oligomeric ethyl ethylene phosphate which can cause an increase in the free volume between the segments of the soft phase. As a result, the amount of heat that must be supplied to change the soft phase temperature by one unit is lower. The data shown in Figure 11 show that the Δcp * of the ZO, EG, and APP foams is significantly greater than for the rest of the foams. Probably, these fillers limited the movement of the flexible segments of the macromolecules of the foam more than others (ATH and MTH). The introduction of keratin to VE + 10 K foam results in increased order-disorder transformation temperature and enthalpy of this transformation compared to VE foam. This indicates that keratin has been chemically bonded to the rigid segments, causing them to stiffen [67]. It has been observed that the order-disorder transformation temperature for ZO, EG, and APP foams is much higher than for other foams. The higher Tt confirms that this group of fillers limits to a greater extent the movement of segments of flexible macromolecules.

One of the thermal analysis techniques used to assess the susceptibility of polymeric materials to thermal degradation is thermogravimetric analysis. For the tested materials, TGA analysis in nitrogen was performed (Figure 12).

The tested VEPUR foams exhibit the typical two-step decomposition behavior under nitrogen atmosphere, representing the decomposition of diisocyanate hard segments and the original polyol—flexible segments. Two or three stages of degradation were observed in the tested materials. There are two stages of degradation in EG foams and three in other foams. On the basis of the Tg curves, the temperature at which a 5% weight loss occurs (Tn5%) and the residue after degradation at 900 °C (Rn900) were determined. The weight loss at each of the degradation stages (mn1, mn2, mn3), the temperature at which the maximum degree of degradation (Tn1, Tn2, and Tn3) is reached, and the maximum rate of degradation (Vn1, Vn2, and Vn3) were determined from the DTG curves. A summary of these parameters is presented in Table 5.

The introduction of keratin filler causes a slight reduction in temperature of 5% weight loss. For all keratin foams but without Fyrol Tn5% it is 249 ÷ 254 °C, and for Fyrol foams it is 233 ÷ 236 °C. The decrease in Tn5% after the incorporation of Fyrol indicates that more volatile substances are released from these foams. After the degradation process at 900 °C, 0.3 to 16.6% of the mass remains. The addition of keratin (VE + 10 K) increases the residue by about 4% compared to the VE foam, and the introduction of Fyrol to the VE + 10 K foam causes the increase in residues by the next 5%. Depending on the flame retardant additionally introduced to the foams with keratin, the amount of residue is different and ranges from 10.6 ÷ 16.6%.

In the temperature range of 180 ÷ 300 °C in polyurethanes, there is a degradation of the rigid segments (HS) [69]. In the subsequent stages of degradation, the flexible segments (SS) decompose. In the initial stage of degradation, urethane bonds in the polyurethane chain begin to break [29,70] and the degradation begins with urethane bonds with the free C = O and then with the bonded C = O. In the tested materials, the first stage of degradation ends at a temperature of approx. 260 ± 5 °C, in this stage 7 ÷ 10.2% of the sample mass is lost. At this stage of degradation, readily volatile substances and rigid segments composed of urethane groups are decomposed. In the second stage, ending at 310 ± 5 °C, 9.5 to 22.5% of the sample weight is lost. During this step, rigid segments composed of urea groups are broken down.

In the case of EG foams, the rigid segments of urea and urethane groups are mixed, which means that in the temperature range up to 320 °C there is one peak related to the degradation of the rigid segments.

The temperature of the maximum rate of HS degradation of urethane group is about 256 °C, and the maximum rate of degradation is about 0.13 ÷ 0.17%/°C, and for urea group it is about 298÷306 °C with a degradation rate of about 0. 24 ÷ 0.46%/°C (Figure 13).

The theoretical proportion of rigid segments in foams is greater than that determined during the TGA analysis. During the degradation at temperatures up to 320 °C, the rigid segments partially decompose, while the aromatic fragments of these segments degrade at temperatures above 400 °C.

In the temperature range of 310 ÷ 415 °C, the degradation of SS occurs. A weight loss of 48 ÷ 68% was observed at the maximum degradation rate of 1.24 ÷ 3.03%/°C and a temperature of 352 ÷ 388 °C.

The theoretical proportion of SS in the foams is greater than that determined on the base of TGA analysis. During the degradation at temperatures up to 415 °C, the flexible segments partially decompose, the residues from the degradation of these segments degrade at higher temperatures (Figure 14).

To analyze the susceptibility to burning of the produced foams, an analysis was carried out using TGA in the air atmosphere (Figure 15). The results of the TG and DTG curve analysis are presented in Table 6.

For a significant number of samples tested in air at temperatures up to 650 °C, four stages of degradation are visible on DTG curves and in the case of VE, VE + 10 K + 10 EG and VE + 10 K + 10 EG + 5 F foams only three stages are visible. The description of the subsequent stages of degradation is summarized in Table 6.

TGA tests in air have shown that the introduction of keratin filler causes a slight reduction in temperature of 5% weight loss for all keratin foams but without Fyrol Ta5% is 242 ÷ 254 °C, and for foams with Fyrol it is 224 ÷ 239 °C. The reduction of Ta5% after the introduction of Fyrol indicates that more volatile substances are released from these foam, similar to the analysis under nitrogen atmosphere. After the degradation process at 900 °C (Ra900), it remains from 0.6 to 12.1% by weight.

The degradation of VE, VE + 10 K + 10 EG, VE + 10 K + 10 EG + 5 F, VE + 10 K + 10 APP, and VE + 10 K + 10 APP + 5 F foams is different from the others.

The first stage of degradation ends at a temperature of about 253 ÷ 262 °C for most foams, and the weight loss in this stage is in the range of 7.1 ÷ 13.8%. At this stage, the maximum degradation rate in the range of 0.21 ÷ 0.31%/°C materials reach at the temperature of 246 ÷ 250 °C. This step can be related to the distribution of rigid segments.

For foams except of VE, VE + 10 K + 10 EG, VE + 10 K + 10 EG + 5 F, VE + 10 K + 10 APP, and VE + 10 K + 10 APP + 5 F in the next two stages of degradation, the mixture of SS and HS decompose. Decomposition occurs in two stages.

The first of these stages ends at a temperature of 333 ÷ 340 °C, and the second at a temperature of 340 ÷ 410 °C. In the first of these stages, Ta2 is approx. 311 ± 5 °C with the degradation rate of approx. 0.66 ± 0.06%/°C, and in the second stage Ta3 is approx. 356 ± 9 °C with the degradation rate of approx. 0.62 ± 0.08%/° C. In the fourth stage of degradation, Ta4 is achieved at the value of approx. 400 ± 10 °C with the degradation rate of approx. 0.22 ± 0.10%/°C.

The introduction of keratin fibers reduces the rate in the second stage of degradation by about 74% (VE + 10 K). Probably keratin fibers limit oxygen transport and diffusion of decomposition products, just like inorganic fillers such as montmoryllonite [71,72].

For most of the tested materials in the air atmosphere, the third stage of degradation ends at a temperature of approx. 380 ÷ 410 °C, in this stage 7 ÷ 10.2% of the sample mass is lost.

The mass loss calorimeter was used to assess the fire hazard of flame-retarded polyurethane foams. As a result, the key parameters, such as average (av-HRR) and the maximum value of heat release rate (pHRR), as well as total heat release (THR), were obtained. Moreover, in Table 7 the percentage mass loss (PML) and pHRR ratio to time of achievement (pHRR/tPHRR), as one of the most popular indices informing about fire growth, were given. Figure 16 presents the representative heat release rate curves as a function of time, recorded during the tests.

The results presented in Figure 16 and Table 7 indicate that most of the series with fire retardants are characterized by the reduced heat release rate, compared to the unmodified polyurethane foam. The only exception was the samples with ZO, which were confirmed by both av-HRR and pHRR values. Similar results of maximum heat release rate to that obtained for VE were observed in the case of VE + 10 K + 5 F, VE + 10 K + 10 MTH and VE + 10 K + 10 MTH + 5 F as well as VE + 10 K + 10 APP + 5 F; however, for this materials the average HRR was significantly reduced. The introduction of keratin caused a decrease in av-HRR by almost half, but its further reduction, as a result of the use of fire retardants, suggesting an additive or synergistic effect, was observed only in the case of foams with ATH and EG. The greatest decrease in HRR values was recorded for expanded graphite (reduction by 75%), for which the lowest pHRR were also noted.

Although only the measure as a function of time, such as HRR curve, contains the full information for comparison as well as to concentrate the appropriate information into a single number, for instance, on the flame spread, different indices have been proposed. Despite the limiting of physical meaning concerning fire behavior, such indices are shown to be superior to pHRR in correlation with other fire scenarios [73]. As can be seen in Table 7, the value of pHRR/tPHRR in the case of unmodified VE reached 1.9 and for many foams with fire retardants was similar or only slightly decreased. A significant reduction was observed only for VE + 10 K + 10 EG as well as VE + 10 K + 10 EG + 5 F and reached 63% and 84%, respectively.

The combination of keratin with also graphite had a positive effect on reducing the THR. In contrast to all other systems, the THR values of VE + 10 K + 10 EG and VE + 10 K + 10 EG + 5 F were reduced by approx. 50% compared to the VE. Total heat release describes a measurement of the fire load of a material, and its decrease shows incomplete combustion, effected either by char formation or by lowered combustion efficiency [74]. Since the yield of percentage mass lost in the case of foams modified with keratin and graphite or keratin, graphite and Fyrol was the lowest and reached 67% and 58%, respectively, the reduction of THR is related mainly to a charring process. Similar behavior during burning caused a decrease in PML values in the case of foams with APP. Moreover, the reduction in THR for foams with ATH and Fyrol indicates incomplete combustion and their gas-phase activity. This effect is known for organophosphorus fire retardants like Fyrol [69]. In turn, ATH decomposes endothermically, causing water released and dilutes the combustible gases [75].

#### 4.2.5. Mechanical Properties

A consequence of the differences in the chemical structure of foams are their properties. The results of their analyzes are presented in Table 8.

After introducing keratin, the hardness of the foams decreases by approx. 37%, and after the introduction of Keratin and Fyrol by 43% (Table 8). All foams with Fyrol are characterized by lower hardness than foams without Fyrol. The hardness of the remaining foams with flame retardant is lower than that of the reference foam, except for the foam with APP-addition.

In industrial practice, the comfort factor (SAG factor) is used to describe the functional properties of foams [76]. The value of the SAG factor depends on the Compression Load Deflection (CLD). At 65% deflection provides the necessary firmness for support and at the initial CLD at 25% deflection gives a luxurious feel. It is assumed that foams with a SAG factor in the range 2–4 are characterized by a high comfort [74]. In the tested foams, CLD25 significantly decreases after the introduction of flame retardant, and CLD65 increases, which causes the SAG factor increase. The decrease in CLD25 is associated with a change in the structure of the foams and the formation of a significant amount of free fragments of cell walls, and the increase in CLD65 is a result of limiting the possibility of macromolecular movement after the introduction of flame retardant (Table 8).

Low compression set of VEPUR is important, e.g., during transport, but also during use. Table 8 summarizes the results of determining the value of compression set for all tested samples. The introduction of flame retardant caused the increased permanent deformation after compression of foams.

## 5. Conclusions

Polyurethane foams belonging to the group of viscoelastic foams can be used in vibration damping devices, but also in existing applications such as mattresses and pillows. In technical devices, materials with increased thermal resistance are sought. In the case of mattresses and pillows produced so far, no focus has been placed on reducing the flammability of such products. However, when these products are used in nursing homes for the long-term sick, the problem of low fire resistance of the products arises as some of the patients are smokers. This is an aging population problem. Despite many works related to reducing the flammability of polyurethane foams, this problem has not been solved. This is especially true of viscoelastic foams, which is why the research presented in the article took up this topic. It was decided to verify the suitability of keratin particles in combination with flame retardant additives for flame retardant viscoelastic foams. Within the work, VEPUR foams with the addition of keratin and other flame retardants were manufactured and characterized. As a result of the research, it was found that the use of these additives causes changes in their chemical structure, especially in the structure of their rigid segments. The introduction of keratin causes an increase in the share of urea groups in HS and an increase in the share of hydrogen bonds connecting HS (Table 3 and Figure 5). The introduction of Fyrol causes changes in the phase structure of foams, resulting in a decrease in the share of urea groups in HS and a decrease in the share of hydrogen bonds connecting HS compared to foams with keratin (Figure 5). This trend continues even after the introduction of other flame retardants. The introduction of keratin causes changes in the cell structure of foam, the size of the pores decreases, and the number of pores increases (Figure 7 and Figure 9). The addition of Fyrol to this foam causes further changes in the pore structure, the amount of damaged pore walls increases significantly, and the pores have more irregular shapes. Keratin is incorporated into the pore walls of foams, and when Fyrol is introduced, the keratin located in the pore walls forms agglomerates (Figure 8). The use of keratin results in an increased apparent density of foam, and its further increase is caused use of Fyrol. This indicates that Fyrol may cause a change during the foam creation process (Table 4).

Thermal analysis of the foams shows that the structure of SS does not change, while the characteristics of the phase being a mixture of SS and HS change. Changes in the temperature of the transformation order-disorder, and the enthalpy of this transformation indicate that keratin was incorporated into polyurethane macromolecules, limiting the possibility of movement of fragments containing this additive (Table 4). On the base of TGA analysis in a nitrogen atmosphere, it was found that the degradation process of the VE, VE + 10 K + 10 EG, VE + 10 K + 10 EG + 5 F, VE + 10 K + 10 APP and VE + 10 K + 10 APP + 5 F foams was different from that of other foams (Table 5). Analysis of the weight loss in the subsequent stages of foam decomposition and their comparison with the theoretical content of different types of foam segments shows that TGA analysis can also be used in extended quantitative analyses. However, this should be confirmed in further systematic studies (Figure 13 and Figure 14).

Thermal analysis of TGA in the air atmosphere confirmed the observation that the process of degradation of the VE, VE + 10 K + 10 EG, VE + 10 K + 10 EG + 5 F, VE + 10 K + 10 APP and VE + 10 K + 10 APP + 5 F foams is different than in the case of other foams (Table 6, Figure 15). The introduction of keratin to VEPUR foam significantly reduces its flammability. The highest fire resistance was achieved for foams containing, apart from keratin, also expanding graphite (Table 7). Research results of the functional properties of the tested foams indicate that the use of flame retardants requires the correction of foam recipes to maintain their functional properties (Table 8).

## Figures and Tables

**Figure 1 polymers-13-01380-f001:**
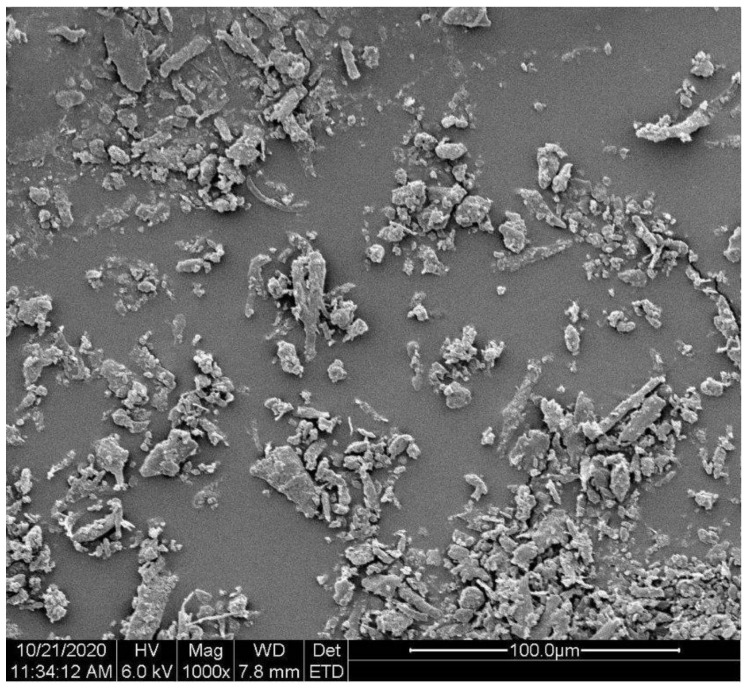
SEM image of the keratin fibers.

**Figure 2 polymers-13-01380-f002:**
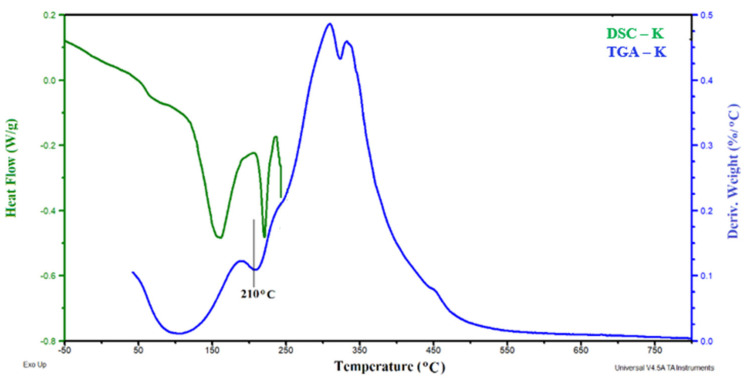
Thermal analysis of keratin (K).

**Figure 3 polymers-13-01380-f003:**
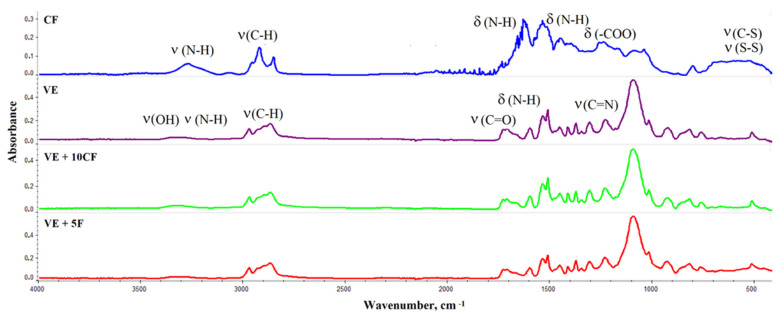
FTIR spectra of K and foams: VE, VE + 10 K and VE + 5 F.

**Figure 4 polymers-13-01380-f004:**
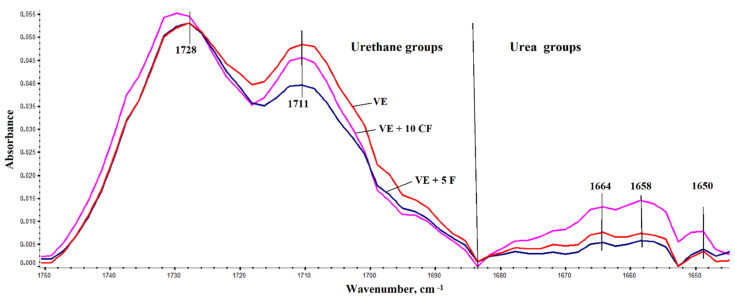
FTIR spectra of polyurethane foams within the range 1750 and 1640 cm^−1^ calibrated with the vibration of the C = C group in the aromatic ring (1596 cm^−1^).

**Figure 5 polymers-13-01380-f005:**
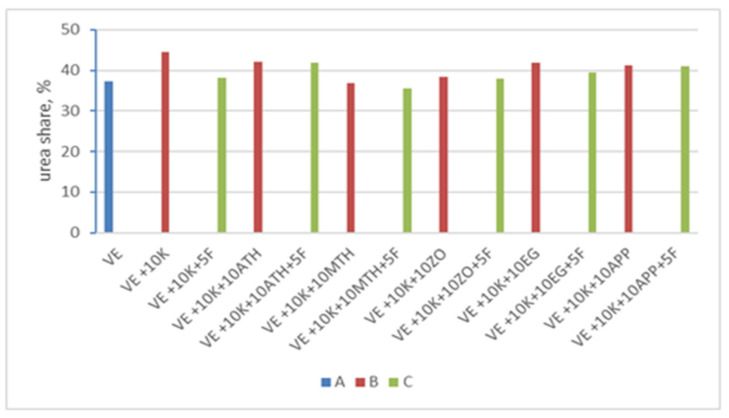
Share of urea bonds in VE foam (**A**), foams with keratin and various flame-retardant additives (**B**), and foams with keratin and various flame-retarding additives and Fyrol (**C**).

**Figure 6 polymers-13-01380-f006:**
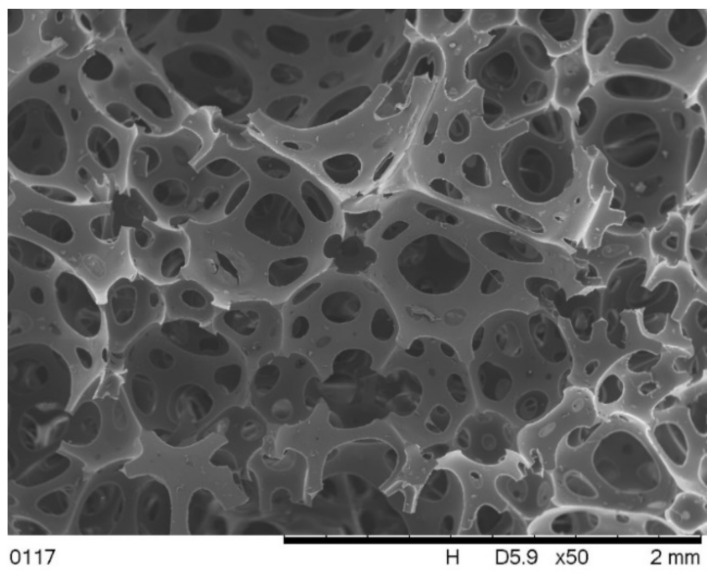
SEM image of the VE foam.

**Figure 7 polymers-13-01380-f007:**
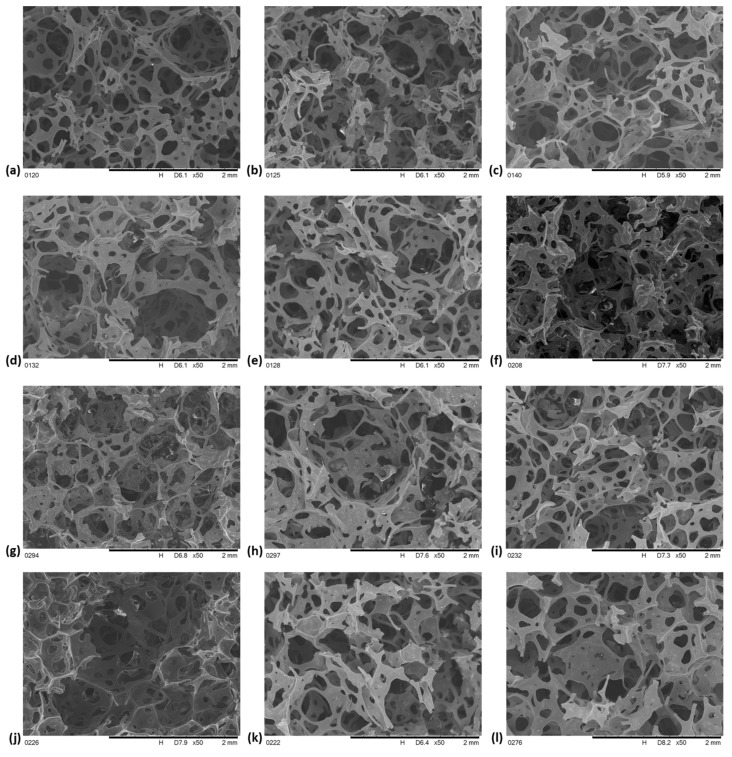
SEM pictures of VEPUR foams with additives: (**a**) VE + 10 K, (**b**) VE + 10 K + 5 F, (**c**) VE + 10 K + 10 ATH, (**d**) VE + 10 K + 10 ATH + 5 F, (**e**) VE + 10 K + 10 MTH, (**f**) VE + 10 K + 10 MTH + 5 F, (**g**) VE + 10 K + 10 ZO, (**h**) VE + 10 K + 10 ZO + 5 F, (**i**) VE + 10 K + 10 EG, (**j**) VE + 10 K + 10 EG + 5 F, (**k**) VE + 10 K + 10 APP, (**l**) VE + 10 K + 10 APP + 5 F.

**Figure 8 polymers-13-01380-f008:**
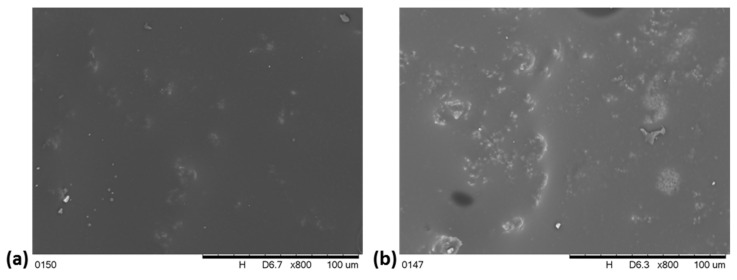
Images of the pore walls in foams: (**a**) VE + 10 K, (**b**) VE + 10 K + 5 F.

**Figure 9 polymers-13-01380-f009:**
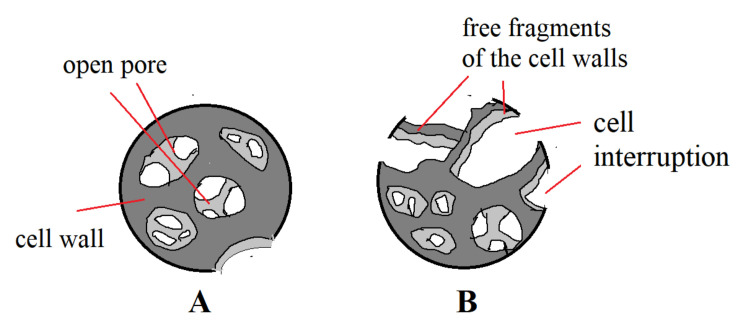
Pore shape diagrams typical for foams with keratin and different flame retardants (**A**) and foams with keratin, different flame retardants and with Fyrol (**B**).

**Figure 10 polymers-13-01380-f010:**
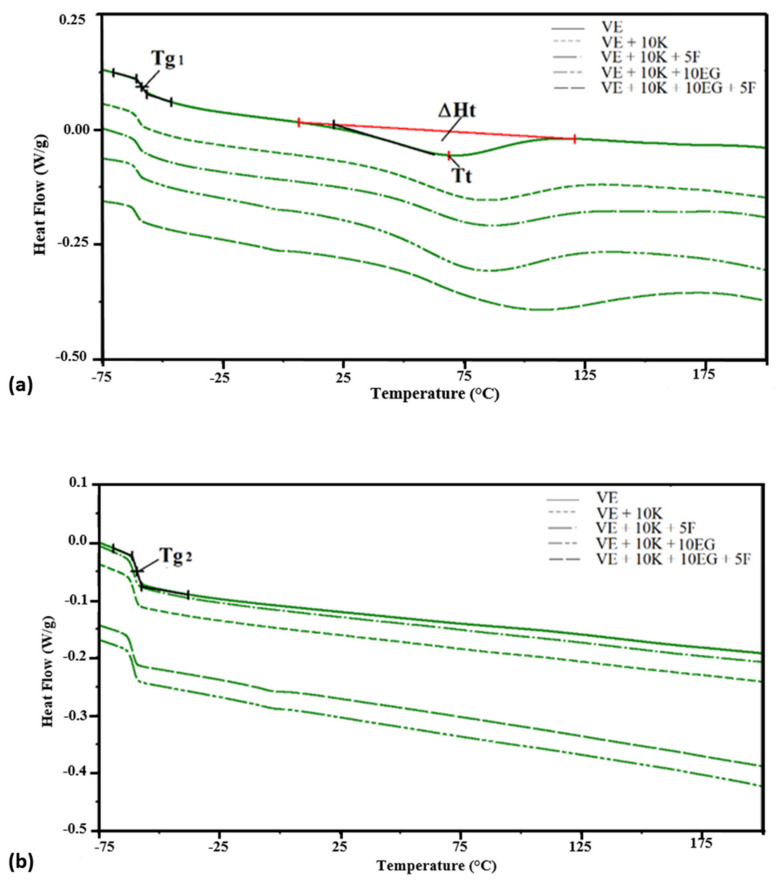
DSC thermograms of selected foams: (**a**) Change of heat flow in the first heating cycle, (**b**) change of heat flow in the second heating cycle.

**Figure 11 polymers-13-01380-f011:**
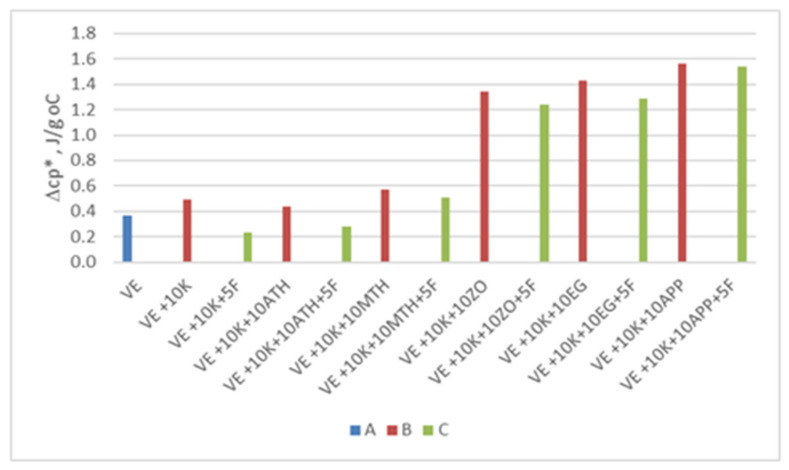
Change of specific heat (Δcp *) in VE foam (**A**), foams with keratin and various flame-retardant additives (**B**) and foams with keratin and various flame-retardant additives and Fyrol (**C**).

**Figure 12 polymers-13-01380-f012:**
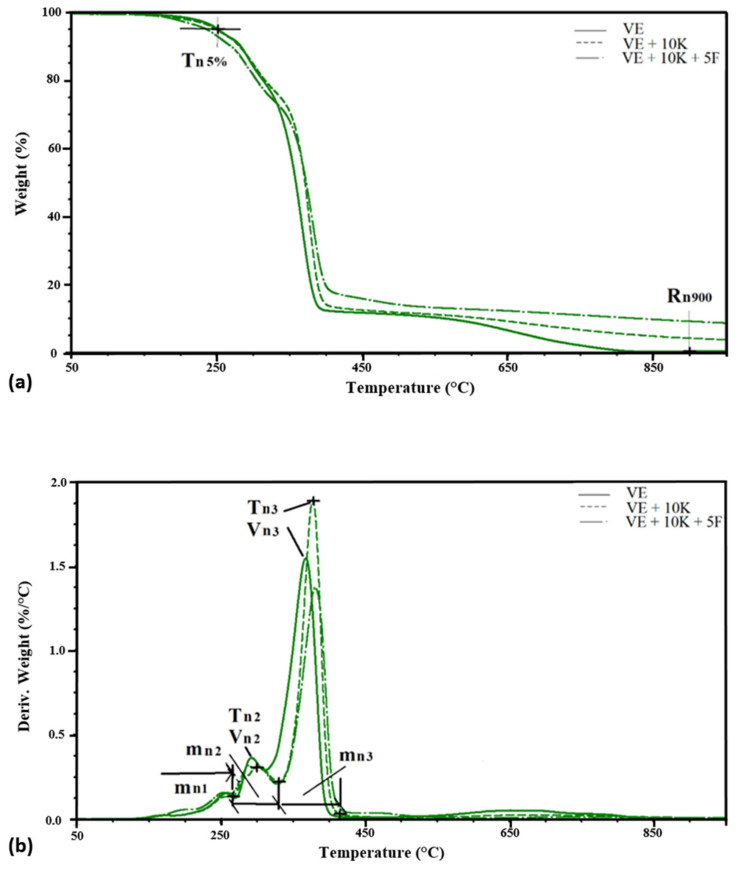
Thermograms of (**a**) TG and (**b**) DTG foams under nitrogen atmosphere.

**Figure 13 polymers-13-01380-f013:**
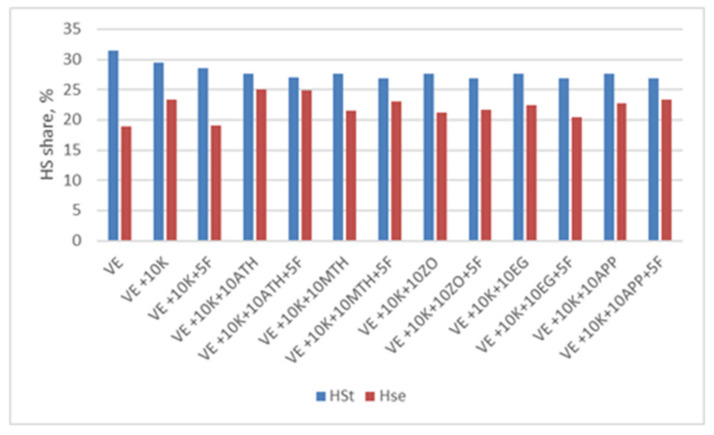
Comparison of the share of HS in the theoretical foam (HSt) and determined from TGA analysis (HSe).

**Figure 14 polymers-13-01380-f014:**
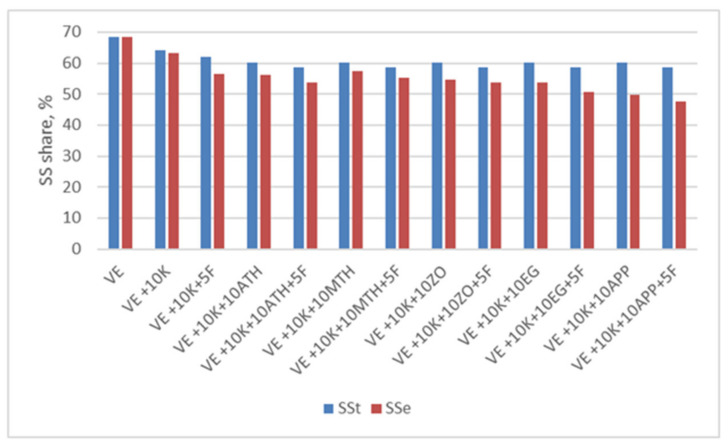
Comparison of the share of SS in the theoretical foam (SSt) and determined from TGA analysis (SSe).

**Figure 15 polymers-13-01380-f015:**
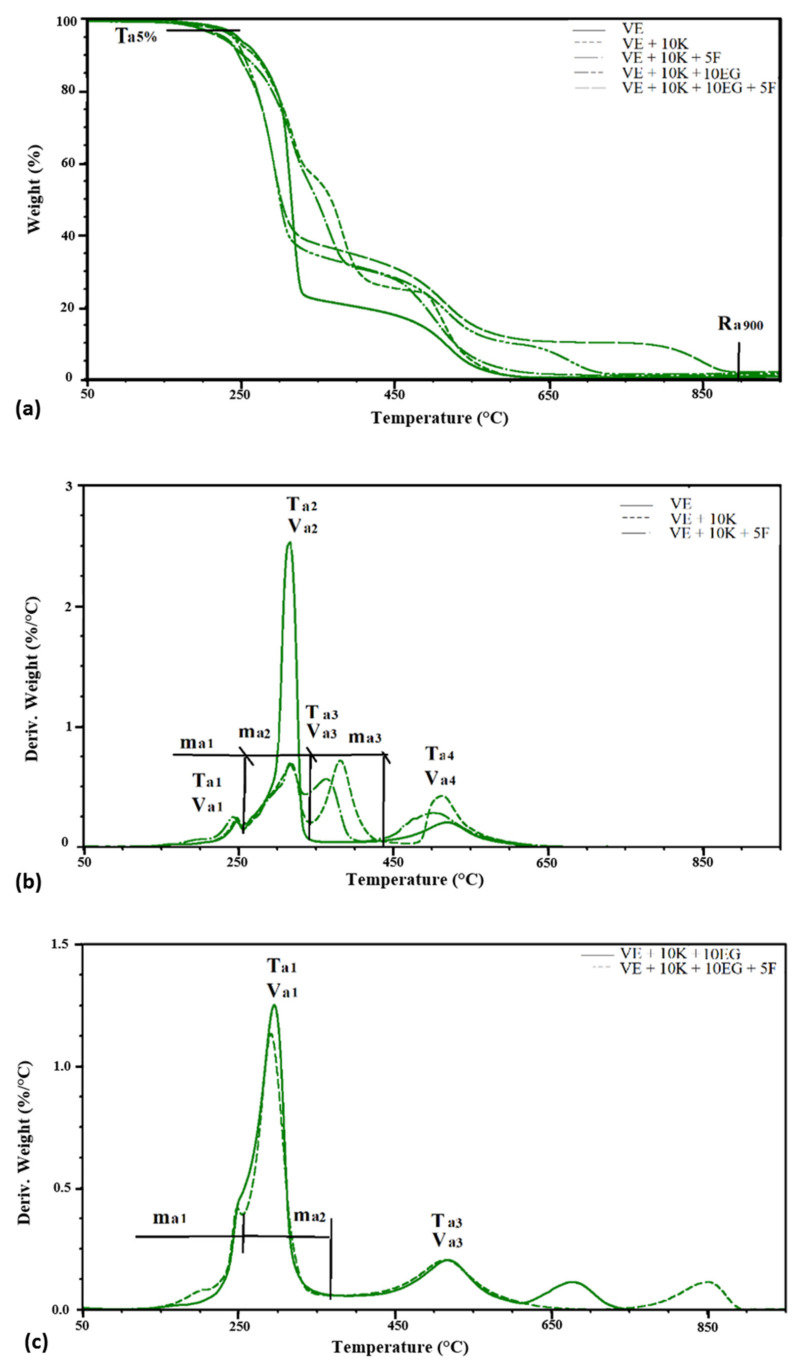
Thermograms of (**a**) TG; (**b**) and (**c**) DTG foams under air atmosphere.

**Figure 16 polymers-13-01380-f016:**
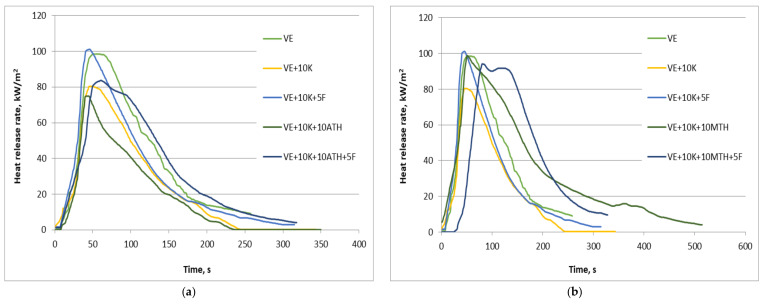
Summary of heat release rate as a function of time for samples: VE, VE + 10 K, VE + 10 K + 5 F, VE + 10 K + 10 ATH, VE + 10 K + 10 ATH + 5 F (**a**); VE, VE + 10 K, VE + 10 K + 5K, VE + 10 K + 10 MTH, VE + 10 K + 10 MTH + 5 F (**b**); VE, VE + 10 K, VE + 10 K + 5K, VE + 10 K + 10 ZO, R + 10 K + 10 ZO + 5 F (**c**); VE, VE + 10 K, VE + 10 K + 5 F, VE + 10 K + 10 EG, R + 10 K + 10 EG + 5 F (**d**); VE, VE + 10 K, VE + 10 K + 5 F, VE + 10 K + 10 APP, VE + 10 K + 10 APP + 5 F (**e**).

**Table 1 polymers-13-01380-t001:** The composition of VEPUR.

Sample VEPUR	Addition	Amount of Additive % mas/Parts per Hundred Parts of Polyol	Component Composition of Foams, wt%		
			Component A	Component B	Sum of Additives
VE *	-	-	69	31	-
VE + 10 K	K	10	64	29	6
VE +10 K +5 F	K	10	62	29	9
	F	5	-	-	-
VE +10 K +10 ATH	K	10	60	28	12
	ATH	10	-	-	-
VE +10 K+ 10 ATH + 5 F	K	10	58	27	15
	ATH	10	-	-	-
	F	5	-	-	-
VE +10 K +10 MTH	K	10	60	28	12
	MTH	10	-	-	-
VE +10 K +10 MTH + 5 F	CF	10	58	27	15
	MTH	10	-	-	-
	F	5	-	-	-
VE +10 K +10 ZO	K	10	60	28	12
	ZO	10	-	-	-
VE +10 K +10 ZO +5 F	K	10	58	27	15
	ZO	10	-	-	-
	F	5	-	-	-
VE +10 K +10 EG	K	10	60	28	12
	EG	10	-	-	-
VE +10 K +10 EG +5 F	K	10	58	27	15
	EG	10	-	-	-
	F	5	-	-	-
VE +10 K +10 APP	K	10	60	28	12
	APP	10	-	-	-
VE +10 K +10 APP +5 F	K	10	58	27	15
	APP	10	-	-	-
	F	5	-	-	-

* VE—VEPUR unmodified foam.

**Table 2 polymers-13-01380-t002:** Description of the band characteristics for the FTIR spectrum of keratin fibers [49,50,51,52,53,54,55].

Wavenumber, cm^−1^	Band Description
3281	Amide A—ν(NH) the 𝛽 or extended polypeptide chain
3071	Amide B—ν𝑎(NH) the 𝛽 or extended polypeptide chain
2928	Asymmetric CH_3_—ν(CH_3_) bands of aliphatic hydrocarbons
2853	Symmetric CH_2_—ν (CH_2_) bands of aliphatic hydrocarbons
1643	Amide I—δ(N-H) bending vibration (of primary amines) of protein molecules
1528	Amide II—δ(N-H) bending vibration (of secondary amines) of protein molecules
1204	Amide III—the valency vibration of δ(-COO) groups
500–670	ν(C-S) and ν(S-S) stretching vibration of sulfur from cysteine of keratin protein

**Table 3 polymers-13-01380-t003:** Characteristics of hard phase of VE and selected VE’s with flame-retardant additives.

VEPUR Foams	Urea Share, %	R	DPS
VE	37.3	1.25	0.55
VE + 10 K	44.6	1.40	0.58
VE + 10 K + 5 F	38.2	1.24	0.55
VE + 10 K + 10 ATH	42.1	1.39	0.58
VE + 10 K + 10 ATH + 5 F	41.9	1.33	0.57
VE + 10 K + 10 MTH	36.9	1.40	0.58
VE + 10 K + 10 MTH + 5 F	35.5	1.25	0.56
VE + 10 K + 10 ZO	38.5	1.35	0.57
VE + 10 K + 10 ZO + 5 F	38.0	1.20	0.55
VE + 10 K + 10 EG	41.8	1.39	0.58
VE + 10 K + 10 EG + 5 F	39.5	1.32	0.57
VE + 10 K + 10 APP	41.3	1.39	0.58
VE + 10 K + 10 APP + 5 F	41. 0	1.25	0.56

**Table 4 polymers-13-01380-t004:** Characteristics of foams: Apparent density and DSC analysis results of VE and VE with additives.

VEPUR	Apparent Density ^a^, kg/m^3^	Tgs1, °C	Δcp *, J/g°C	Tt, °C	ΔHt, J/g	Tgs2, °C
VE	55.5 (0.8)	−58.6	0.37	18.8	68.2	−59.2
VE + 10 K	61.5 (1.8)	−60.0	0.49	19.1	80.1	−60.4
VE + 10 K + 5 F	69.3 (1.2)	−60.2	0.32	20.6	81.6	−60.5
VE + 10 K + 10 ATH	73.6 (1.2)	−59.9	0.44	17.4	75.2	−60.7
VE + 10 K + 10 ATH + 5 F	80.2 (1.7)	−60.1	0.28	17.6	79.2	−60.2
VE + 10 K + 10 MTH	83.5 (2.2)	−60.0	0.54	17.4	77.7	−60.3
VE + 10 K + 10 MTH + 5 F	95.1 (4.1)	−60.2	0.31	17.5	76.9	−60.2
VE + 10 K + 10 ZO	82.4 (4.3)	−60.2	1.34	32.7	76.5	−60.8
VE + 10 K + 10 ZO + 5 F	83.8 (1.5)	−60.5	1.24	27.2	78.2	−61.2
VE + 10 K + 10 EG	71.3 (1.8)	−60.7	1.43	28.1	76.7	−60.8
VE + 10 K+10 EG+5 F	79.0 (2.4)	−60.9	1.29	34.3	81.8	−61.1
VE + 10 K+10 APP	88.0 (1.8)	−60.7	1.56	29.1	76.3	−61.0
VE + 10 K+10 APP+5 F	87.6 (2.3)	−61.0	1.44	34.0	82.2	−61.0

^a^ The values in parentheses refer to standard deviations.

**Table 5 polymers-13-01380-t005:** Results of TGA analysis under nitrogen (*n*).

VEPUR	Tn5%, C	mn1, g	mn2, g	mn3, g	Rn900, %	Tn1, °C	Vn1, %/°C	Tn2, °C	Vn2, %/°C	Tn3, °C	Vn3, %/°C
VE	252	7.5	11.4	68.4	0.3	256	0.15	292	0.36	368	1.55
VE + 10 K	249	7.3	16.0	63.3	4.3	256	0.14	302	0.31	377	1.91
VE + 10 K + 5 F	236	9.5	9.5	56.5	9.2	256	0.16	303	0.31	381	1.38
VE + 10 K + 10 ATH	248	7.0	18.1	56.2	11.3	256	0.14	297	0.43	375	1.60
VE + 10 K + 10 ATH + 5 F	236	9.4	16.5	53.8	12.6	257	0.19	292	0.46	379	1.24
VE + 10 K + 10 MTH	254	6.2	15.3	57.4	12.1	256	0.13	300	0.30	381	1.59
VE + 10 K + 10 MTH + 5 F	241	9.0	14.0	55.2	13.8	261	0.15	306	0.26	388	1.36
VE + 10 K + 10 ZO	252	7.3	13.9	54.7	14.3	257	0.14	298	0.29	380	1.68
VE + 10 K + 10 ZO + 5 F	240	9.1	12.5	53.6	14.7	258	0.15	306	0.24	388	1.35
VE + 10 K + 10 EG	245	nd	22.5	53.8	10.6	nd	nd	285	0.33	372	1.51
VE + 10 K + 10 EG + 5 F	228	nd	20.4	50.8	14.0	nd	nd	286	0.30	375	1.21
VE + 10 K + 10 APP	249	7.4	15.4	49.8	14.9	256	0.16	299	0.36	352	3.03
VE + 10 K + 10 APP + 5 F	233	10.2	13.2	47.7	16.6	256	0.17	295	0.32	352	2.95

nd—not determined.

**Table 6 polymers-13-01380-t006:** Results of TGA analysis under air (a).

VEPUR	Ta5%, °C	ma1, %	ma2, %	ma3, %	Ra900, %	Ta1, °C	Va1, %/°C	Ta2, °C	Va2, %/°C	Ta3, °C	Va3, %/°C	Ta4, °C	Va4, %/°C
VE	246	7.1	70.1	nd	0.6	248	0.21	316	2.53	nd	nd	nd	nd
VE + 10 K	242	8.3	34.9	32.2	0.8	248	0.24	318	0.69	382	0.72	514	0.42
VE + 10 K + 5 F	228	10.0	34.4	25.2	1.1	245	0.25	316	0.69	363	0.57	503	0.28
VE + 10 K + 10 ATH	242	9.8	42.6	17.1	8.0	248	0.24	298	0.59	346	0.54	507	0.21
VE + 10 K + 10 ATH + 5 F	231	10.6	39.8	15.5	8.5	246	0.31	291	0.57	318	0.58	509	0.22
VE + 10 K + 10 MTH	245	7.5	40.8	19.8	8.6	250	0.22	307	0.69	361	0.51	497	0.19
VE + 10 K + 10 MTH + 5 F	235	10.4	33.5	23.6	8.9	248	0.25	323	0.68	365	0.66	514	0.20
VE + 10 K + 10 ZO	245	8.2	41.9	14.5	12.1	250	0.24	308	0.67	355	0.55	491	0.23
VE + 10 K + 10 ZO + 5 F	236	10.5	34.0	18.7	12.1	248	0.28	323	0.63	356	0.59	511	0.22
VE + 10 K + 10 EG	239	13.8	50.9	nd	1.5	250	0.45	294	1.25	nd	nd	517	0.22
VE + 10 K + 10 EG + 5 F	224	13.3	48.7	nd	2.0	247	0.42	291	1.13	nd	nd	518	0.20
VE + 10 K + 10 APP	242	7.8	12.2	40.9	7.8	246	0.25	280	0.39	328	1.19	486	0.13
VE + 10 K + 10 APP + 5 F	229	9.7	11.7	40.2	7.5	250	0.23	280	0.38	333	1.07	488	0.13

**Table 7 polymers-13-01380-t007:** The cone calorimeter results of VE and VE with fire retardant.

Sample	av-HRR, kW/m^2^	pHRR, kW/m^2^	pHRR/t_PHRR_, kW/m^2^s^1^	THR, MJ/m^2^	PML, %
VE	75 (5)	104 (8)	1.9 (0)	7.9 (0)	94.5 (0)
VE + 10 K	40 (6)	81 (1)	1.7 (0)	7.0 (1)	94.2 (4)
VE + 10 K + 5 F	58 (14)	106 (7)	1.7 (1)	10.0 (1)	89.3 (1)
VE + 10 K + 10 ATH	36 (11)	94 (27)	1.4 (0)	9.0 (4)	87.7 (4)
VE + 10 K + 10 ATH + 5 F	50 (3)	87 (4)	1.4 (0)	8.4 (2)	83.4 (1)
VE + 10 K + 10 MTH	50 (10)	101 (3)	2.0 (0)	14.3 (3)	88.6 (3)
VE + 10 K + 10 MTH + 5 F	60 (12)	101 (10)	1.4 (0)	12.9 (0)	83.8 (1)
VE + 10 K + 10 ZO	71 (12)	110 (10)	1.5 (0)	12.1 (1)	83.5 (1)
VE + 10 K + 10 ZO + 5 F	92 (41)	156 (36)	2.0 (1)	11.1 (1)	80.8 (1)
VE + 10 K + 10 EG	19 (4)	31 (2)	0.7 (1)	4.0 (0)	67.3 (12)
VE + 10 K + 10 EG + 5 F	24 (3)	30 (6)	0.3 (0)	4.5 (1)	58.4 (3)
VE + 10 K + 10 APP	55 (11)	92 (13)	1.3 (0)	10.9 (3)	81.0 (0)
VE + 10 K + 10 APP + 5 F	66 (17)	107 (13)	1.9 (0)	9.2 (0)	79.2 (3)

The values in parentheses refer to standard deviations.

**Table 8 polymers-13-01380-t008:** Static properties of foams.

VEPUR	CLD 40. kPa	SAG	Compression by 90%	Compression by 75%	Compression by 50%
			The Compression set. %		
VE	4.9 (0.3)	2.9 (0.2)	14 (3)	10 (3)	10 (3)
VE + 10 K	3.1 (0.2)	4.8 (0.4)	25 (9)	16 (5)	17 (3)
VE + 10 K + 5 F	2.8 (0.3)	6.6 (0.5)	24 (8)	18 (4)	17 (4)
VE + 10 K + 10 ATH	3.9 (0.3)	5.3 (0.5)	57 (11)	23 (6)	16 (3)
VE + 10 K + 10 ATH + 5 F	3.1 (0.2)	6.5 (0.5)	39 (4)	22 (5)	20 (4)
VE + 10 K + 10 MTH	4.5 (0.3)	8.9 (0.7)	55 (9)	33 (6)	27 (4)
VE + 10 K + 10 MTH + 5 F	3.4 (0.3)	8.2 (0.8)	51 (8)	24 (3)	20 (3)
VE + 10 K + 10 ZO	4.2 (0.4)	7.5 (0.6)	62 (12)	45 (3)	24 (4)
VE + 10 K + 10 ZO + 5 F	3.1 (0.2)	7.8 (0.6)	43 (5)	48 (4)	25 (3)
VE + 10 K + 10 EG	3.8 (0.3)	6.7 (0.5)	59 (6)	56 (9)	20 (3)
VE + 10 K + 10 EG + 5 F	3.0 (0.3)	8.7 (0.7)	54 (5)	33 (5)	26 (5)
VE + 10 K + 10 APP	5.6 (0.4)	6.2 (0.5)	37 (4)	22 (3)	17 (3)
VE + 10 K + 10 APP+5 F	3.9 (0.4)	7.3 (0.5)	50 (7)	34 (5)	25 (5)

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
