# Peer review of "Viscoelastic Polyurethane Foam with Keratin and Flame-Retardant Additives"

_polymers, 2021, doi:10.3390/polym13091380_

Round 1
Reviewer 1 Report
The authors reported that: "Polyurethane foams belonging to the group of viscoelastic foams can be used in vibration damping devices, but also in existing applications such as mattresses and pillows. For this reason, the mechanical tests need to be integrated with stress-strain curves in order to understand the effective role of the keratin fibres in the foams.
-Moreover, thermal measurements (conductivity, diffusivity) must be reported in order to show the insulation properties of these materials as a function of the different compositions.
The apparent density values are not enough to characterize the different types of foams. Porosity measurement are needed, specifically pore size and pore size distribution.
Error bars must be reported in the figures.
The authors must report the effective novelty of this work with respect to what reported in literature.
Major revisions are required for this paper
Author Response
- The authors reported that: "Polyurethane foams belonging to the group of viscoelastic foams can be used in vibration damping devices, but also in existing applications such as mattresses and pillows. For this reason, the mechanical tests need to be integrated with stress-strain curves in order to understand the effective role of the keratin fibres in the foams.
We agree with the comment of the reviewer that it is beneficial to present the stress-strain curves for the materials tested in this article in order to demonstrate their viscoelastic properties. Table 8 presents the results obtained from the analysis of the stress-strain curves (CLD 40 and SAG factor). The results of these studies are interpreted in lines 623-641. In subsequent works on foams with keratin fibers, we will focus on the research aspect proposed by the Reviewer. In the presented research, we focused on the flammability of foams. More research results related to the features of this group of foams are presented in other recent and earlier articles by co-authors, e.g .:
Okrasa, M.; Leszczyńska, M.; Sałasińska, K.; Szczepkowski, L.; Kozikowski, P.; Majchrzycka, K.; Ryszkowska, J. Viscoelastic Polyurethane Foams for Use in Seals of Respiratory Protective Devices. Materials 2021, 14, 1600. doi: 10.3390/ma14071600
AuguÅ›cik-Królikowska, M.; Ryszkowska, J.; KuraÅ„ska, M.; Wantulok, M.; Gloc, M.; Szczepkowski, L.; DÄ…bkowska-SusfaÅ‚, K.; Prociak, A. Composites of Open-Cell Viscoelastic Foams with Blackcurrant Pomace. Materials 2021, 14, 934. doi: 10.3390/ma14040934
- Moreover, thermal measurements (conductivity, diffusivity) must be reported in order to show the insulation properties of these materials as a function of the different compositions.
Usually, thermal measurements (conductivity, diffusivity) are carried out to demonstrate thermal insulation properties of foams. Such tests are performed on rigid foams, not flexible foams, which include viscoelastic foams. Therefore, this aspect of research is not included in the article.
- The apparent density values are not enough to characterize the different types of foams. Porosity measurement are needed, specifically pore size and pore size distribution.
We agree with the Reviewer that apart from the apparent density, it is good to assess the porosity of the foams. If it is possible, we try to take into account such research, for example in the work: AuguÅ›cik-Królikowska, M .; Ryszkowska, J .; KuraÅ„ska, M .; Wantulok, M .; Gloc, M .; Szczepkowski, L .; DÄ…bkowska-SusfaÅ‚, K .; Prociak, A. Composites of Open-Cell Viscoelastic Foams with Blackcurrant Pomace. Materials 2021, 14, 934. (doi: 10.3390/ma14040934). The best technique for testing this group of foams is microtomography, in the case of foams from the tested group, this test may be burdened with very large errors. These errors may result from the use of more additives used in their production. To complete the characteristics of the materials, the article presents SEM images of foams.
- Error bars must be reported in the figures.
We agree with the comment of the Reviewer on measurement errors. Unfortunately, the figures compare the test results obtained on the basis of one or two DSC and TGA measurements. The small number of measurements resulted from the limited costs of their realisation. The comparisons presented in the figures show that it is worth using the results of thermal analyzes in this way. Therefore, the authors, in further work on similar groups of materials, will carry out more DSC and TGA measurements.
- The authors must report the effective novelty of this work with respect to what reported in literature.
So far, the literature has not presented the results of tests on mixtures of flame retardant additives with keratin for flame retardant viscoelastic foams. It is very difficult to achieve high fire resistance of this group of foams, therefore the search for new solutions for this group of materials is the current research trend. The authors presented this information in the introduction to the article.

Reviewer 2 Report
The present study reports on viscoelastic polyurethane foam with keratin and flame-retardant additives, and a series of experimental results have been carried out to support it. After carefully reading it, the following points are suggested to consider to improve this study.
1. In Abstract and Introduction sections, “viscoelastic polyurethane foams (VEPUR)” is revised to “viscoelastic polyurethane (VEPUR) foams”
2. What about the “flexible foams (FPUF)”? How the FPUF is used for flexible polyurethane foams? Please check it.
3. The introduction is so long, it should be shortened. The logic is not so clear.
4. For “The foams were made using the following flame retardant additives”, a table is suggested to use.
5. “3. Research Metodology” is revised to “Experimental”; while the word of Metodology is not correct.
6. In section of 4.1, “The chemical structure of K fibres was analyzed using infrared spectroscopy, the FTIR spectrum is shown in Fig. 3. The results of this analysis are summarized in Table 2.” Should be removed to “4.3”.
7. Section 4.2 is removed to section of “3. Research Metodology”.
8. The quality of Figures 10, 15 and 16 should be improved.
9. The discussion on the working principle of keratin on VEPUR foams is not enough, whether the keratin has a high thermal resistivity property? Or the keratin is able to transfer the heat for foams? It is difficult to catch up with it. The following references can be cited for improving the discussion on it. i.e., (1) Andong Liu, Andreas Walther, Olli Ikkala, Lyuba Belova, Lars A Berglund. Clay nanopaper with tough cellulose nanofiber matrix for fire retardancy and gas barrier functions. Biomacromolecules. 2011, 12(3):633-41. (2) Haibao Lu, Fei Liang and Jihua Gou. Nanopaper Enabled Shape-Memory Nanocomposite with Vertically Aligned Nickel Nanostrand: Controlled Synthesis and Electrical Actuation. Soft Matter. 2011, 7(16): 7416-7423. (3) F Carosio , J Kochumalayil, F Cuttica, G Camino, L Berglund. Oriented clay nanopaper from biobased components--mechanisms for superior fire protection properties. ACS Appl Mater Interfaces. 2015, 7(10):5847-56. (4) Haibao Lu and Wei Min Huang. Synergistic effect of self-assembled carboxylic acid-functionalized carbon nanotubes and carbon fiber for improved electro-activated polymeric shape-memory nanocomposite. Applied Physics Letters. 2013, 102(23): 231910.
10. The experimental result of viscoelastic properties of foams has not been presented and discussed.
In all, the present study should be carefully revise before recommendation for publication.
Author Response
The present study reports on viscoelastic polyurethane foam with keratin and flame-retardant additives, and a series of experimental results have been carried out to support it. After carefully reading it, the following points are suggested to consider to improve this study.
- In Abstract and Introduction sections, “viscoelastic polyurethane foams (VEPUR)” is revised to “viscoelastic polyurethane (VEPUR) foams”
Corrected according to the Reviewer's note.
- What about the “flexible foams (FPUF)”? How the FPUF is used for flexible polyurethane foams? Please check it.
The nomenclature has been arranged according to the note.
- The introduction is so long, it should be shortened. The logic is not so clear.
Introduction is shortened.
- For “The foams were made using the following flame retardant additives”, a table is suggested to use.
The description of the applied additives has been corrected
- “3. Research Metodology” is revised to “Experimental”; while the word of Metodology is not correct.
Chapter title changed.
- In section of 4.1, “The chemical structure of K fibres was analyzed using infrared spectroscopy, the FTIR spectrum is shown in Fig. 3. The results of this analysis are summarized in Table 2.” Should be removed to “4.3”.
The article proposes a division into two sections, the first one describes the filler and the second one describes foams and their modifications. In order to remain consistent in this division, the numbering and titles of the subsections have been changed and rearranged.
- Section 4.2 is removed to section of “3. Research Metodology”.
The numbering of subsections has been changed and reorganized.
- The quality of Figures 10, 15 and 16 should be improved.
The authors improved the quality of Figures 10, 12, 15 and 16
- The discussion on the working principle of keratin on VEPUR foams is not enough, whether the keratin has a high thermal resistivity property? Or the keratin is able to transfer the heat for foams? It is difficult to catch up with it. The following references can be cited for improving the discussion on it. i.e., (1) Andong Liu, Andreas Walther, Olli Ikkala, Lyuba Belova, Lars A Berglund. Clay nanopaper with tough cellulose nanofiber matrix for fire retardancy and gas barrier functions. Biomacromolecules. 2011, 12(3):633-41. (2) Haibao Lu, Fei Liang and Jihua Gou. Nanopaper Enabled Shape-Memory Nanocomposite with Vertically Aligned Nickel Nanostrand: Controlled Synthesis and Electrical Actuation. Soft Matter. 2011, 7(16): 7416-7423. (3) F Carosio , J Kochumalayil, F Cuttica, G Camino, L Berglund. Oriented clay nanopaper from biobased components--mechanisms for superior fire protection properties. ACS Appl Mater Interfaces. 2015, 7(10):5847-56. (4) Haibao Lu and Wei Min Huang. Synergistic effect of self-assembled carboxylic acid-functionalized carbon nanotubes and carbon fiber for improved electro-activated polymeric shape-memory nanocomposite. Applied Physics Letters. 2013, 102(23): 231910.
In the discussion of the results, the authors cited 2 of the articles proposed by the Reviewer.
- The experimental result of viscoelastic properties of foams has not been presented and discussed.
In the case of viscoelastic foams, the parameters presented in Table 8 are usually used to characterize their viscoelastic properties. These results are discussed below the table.

Round 2
Reviewer 1 Report
The paper can be now accepted for publication
This manuscript is a resubmission of an earlier submission. The following is a list of the peer review reports and author responses from that submission.
Round 1
Reviewer 1 Report
Dear Editor and Authors,
I have never seen a manuscript prepared with such negligence for second review before. The reviewer does not understand why the text of the manuscript differs in font size and type. The lack of visual design in the arrangement of the measurement data, the absence of several graphs of data at all.
As it stands, this manuscript is not suitable for peer review, nor any attempt toward publication!!!
Authors must prepare the manuscript carefully and then send it for Review.
Reviewer 2 Report
I have gone through the manuscript Viscoelastic polyurethane foam with keratin and flame-retardant additives
There are many English and grammar corrections. It looks like a master thesis not like a research paper. The present form of the manuscript is not in the consistent flow which is difficult to understand and to identify the actual theme of the research work.
The concept and research work is interesting however the ways of presentation is not good. Thus authors should thoroughly work and modify the whole content of the manuscript. After substantial revision authors are encouraged to resubmit the improved version of the manuscript. In the present form it can not be accepted for publication.